# Osteichthyan Fishes from the uppermost Norian (Triassic) of the Fuchsberg near Seinstedt, Lower Saxony (Germany)

**Hans-Peter Schultze [1],\*, Gloria Arratia [1], Norbert Hauschke [2] and Volker Wilde [3]**

1 Biodiversity Institute, University of Kansas, Dyche Hall, 1345 Jayhawk Blvd., Lawrence, KS 66045, USA
2 Institut für Geowissenschaften und Geographie, Martin-Luther-Universität Halle-Wittenberg, Von-Seckendorff-Platz 3, D-06120 Halle (Saale), Germany
3 Senckenberg Gesellschaft für Naturforschung, Senckenberganlage 25, D-60325 Frankfurt am Main, Germany
\* Correspondence: hp1937@ku.edu

**Abstract:** Fishes from the uppermost Norian Fuchsberg Quarry near Seinstedt are represented by two taxa that we interpret as a teleosteomorph (complete specimens) and actinistian (scales). *Seinstedtia parva* gen. et sp. nov. is described; although it was proposed as a possible semionotiform, this study reveals that *Seinstedtia* possesses a combination of teleosteomorph features, for instance: characteristic pholidophoriform-shaped cranial roof; fusion of skull roof bones; three dorso-posterior infraorbitals, including an enlarged infraorbital 3; one suborbital bone; movable premaxilla; and characteristic-shaped preopercle. In parallel, *Seinstedtia* possesses a head gently curved anteriorly, with lower jaw protruding slightly in front of upper jaw; supraorbital 1 forming most of antero-dorsal margin of circumorbital ring; one supramaxilla; three extrascapulae; cleithrum with short and broad lower arm; and enlarged clavicle. This character combination places *Seinstedtia* as a teleosteomorph, family incertae sedis. This fish (total length ca. 50 mm) and some pholidophorids (ca. 70 mm or less; *Parapholidophorus nybelini* and *Pholidoctenus serianus*) represent the known smallest teleosteomorphs that inhabited Europe during the Norian. The isolated scales are elasmoid of amioid type ornamented with the elongated ridges of actinistians. This diversity of teleosteomorphs and actinistians in Fuchsberg Quarry during the Triassic indicates a connection to a marine environment.

**Keywords:** Actinistia; Actinopterygii; Teleosteomorpha; Late Triassic; taxonomy; morphology; miniaturization

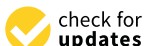



## 1. Introduction

The upper Keuper ("Rhät") in the surroundings of Braunschweig (Brunsvig) has been studied in some detail e.g., [1–4]. The old quarry at Fuchsberg is situated on the northern flank of Fallstein in a ridge within the upper Keuper between Seinstedt and Hedeper (Lower Saxony, Germany). Fallstein is one of the prominent salt-induced anticlines characterizing the northern foreland of the Harz Mountains [5]. It has Lower Triassic Buntsandstein exposed on its top and is surrounded by younger strata, especially rocks of Muschelkalk and Keuper facies. The Fuchsberg Quarry shows weakly cemented light sandstones with intercalations of light gray to greenish silt and clay stones. Due to a lack of biostratigraphic data, the respective strata have traditionally been assigned to the lower part of the Rhät ("Unterrhät") that has been lithostratigraphically sandwiched between two regionally extended characteristic layers, a bonebed at the base that is no longer exposed in the Fuchsberg Quarry and a highly bioturbated greenish siltstone at the top ("Grüner Grenzhorizont") that is still seen in the upper part of the quarry [3,6].

Fossils have repeatedly been recovered from the Fuchsberg Quarry, including a prosauropod humerus (*Plateosaurus* sp. [7]), some plant remains [3,8–10], conchostracans [3,8,11–15] and diverse insects and other arthropods, including limulids [8–10]. Most of the fossil material including the fish remains has been found in the uppermost part of a 1.33 to 1.44 m thick alternation of sand-, silt-, and claystone layers of cm to dm thickness,

the "Fossilführende Wechselfolge" (fossiliferous strata) of Hauschke and Wilde [13]. Systematic studies of the conchostracans recently revealed biostratigraphically relevant taxa that indicate a late Norian to early Rhaetian age for the Fuchsberg sequence [14,15], but the Fossilführende Wechselfolge obviously still belongs to the Norian [14–16]. The insects include a biogeographically important genus (*Ipsvicia*) that had not been recorded outside of Gondwana until 2011 [17].

Previous sedimentological studies [3,12,18] interpreted the Fuchsberg Quarry and adjacent outcrops in terms of a fluvio-lacustrine or deltaic environment for the upper Norian to lower Rhaetian (Unterrhät) of the area. Due to a lack of bioturbation, rooting and desiccation cracks, the Fossilführende Wechselfolge has been interpreted as deposits of a permanent lake with anoxic conditions at the bottom [13]. A brackish to fluviatile environment is indicated by fossil occurrences and sedimentological studies [16] at the Fuchsberg. The poor plant taphocoenosis in combination with a dominance of gymnosperm pollen and the absence of root traces point to a comparatively dry climate, as suggested by Jüngst [3] for the upper Norian to lower Rhaetian [16].

*The Fish Material*

Barth et al. ([16]: table 4) listed seven chondrichthyans (*Polyacrodus* sp., *Lissodus nodosus*, *Pseudodalatias barnstonensis*, *Synechodus* n. sp. 1 and n. sp. 2, *Rhomphaiodon minor*, *R. nicolensis*, Synechodontiformes n. gen. n. sp., and *Pseudocetorhinus pickfordi*) and five actinopterygians (*Gyrolepis albertii*, *Severnichthys acuminatus*, Semionotiformes-like, *Colobodus* ?sp., and *Serrolepis suevicus*) as well as actinistian scales at the locality. Chondrichthyans and actinopterygians occur in bone beds I (Langenberg 1) and II (north of Fuchsberg main outcrop at Fuchsberg 13), whereas teleosteomorph specimens and coelacanthiform scales are found in horizon MFL at the Fuchsberg main outcrop (section F1).

*Severnichthys acuminatus*, as described by Storrs [19], is a misidentification of *Birgeria acuminata* ([20]: p. 252). Storrs [19] combined the snout of an amphibian with folded teeth with a lower jaw of *Birgeria acuminata*. In Barth et al. [16], an isolated tooth was compared with a single tooth in Storrs ([19]: figure 5C; written communication by D. Thies, September 2021). The fishes previously interpreted as Semionotiformes (either cf. *Semionotus* sp. or *Semionotus*-like fish) are here described and taxonomically reassigned.

## 2. Material and Methods

The fossil material is deposited in the Geowissenschaftliche Sammlungen, Zentralmagazin Naturwisssenschaftlicher Sammlungen, Martin Luther-Universität Halle-Wittenberg, Halle (Saale), Germany (**MLU**). A broad comparison was made with other Triassic teleosteomorphs deposited in the Geological-Palaeontological Section of the Naturhistorical Museum, Vienna, Austria (**NHMW**), Palaeontological Institut and Museum, University of Zurich, Switzerland (**PIMUZ**), and the Civic Museum of Natural Science Enrico Caffi, Bergamo, Italy (**MCSNB**).

### 2.1. Anatomical Terminology

The terminology of the skull roof bones is based on homology and ontogeny ([21,22] and literature cited therein). To avoid confusion, the first time that the parietal and postparietal bones are cited in the text, as well as in figures, the traditional terminology is shown in square brackets, e.g., parietal bone [=frontal]: pa [=fr]. The terms for fin rays, scutes, and fulcra follow Arratia [23,24] and that for urodermals follows Arratia and Schultze [25].

### 2.2. Illustrations

Illustrations are based directly on the specimens. The drawings were prepared with help of a camera lucida attached on a WILD stereomicroscope M5A and a Zeiss 47 50 52 stereomicroscope, and photographs were taken with a Nikon R9 and 30 mm lens.

### 3. Systematic Paleontology

This section is divided in two parts; the first one concerns fishes interpreted as teleosteomorphs, and the second part is dedicated to the actinistian scales.

#### 3.1. Teleosteomorphs

Osteichthyes Huxley, 1880 [26].
Actinopterygii Cope, 1887 [27].
Neopterygii Regan, 1923 [28].
Teleosteomorpha Arratia, 2001 [29].
Family incertae sedis.

#### 3.1.1. *Seinstedtia* gen. nov.

**Diagnosis**: The generic diagnosis is based on a unique combination of characteristics; an asterisk [*] indicates a possible uniquely derived characteristic within teleosteomorphs. Miniature fish of about 50 mm total length. Oblong fish with a slightly sharp anterior profile of head formed by anterior tip of parietal bones, elongate nasals, a small rostral, and small premaxillae. Short snout, about one-tenth head length. Skull roof pholidophorid-like, markedly narrow anteriorly and broadly expanded posteriorly, giving a characteristic triangular contour. Parietal [=frontal], postparietal [=parietal], and dermopterotic partially to completely fused. One median plus two lateral extrascapulae [*]. Lower jaw slightly projected in front of the upper jaw. First supraorbital forming most of the antero-dorsal corner of the circumorbital ring [*]. Enlarged infraorbital 3 [=jugal] with length equal to length of posterior infraorbital plus length of suborbital that extends to the anterior margin of preoperculum. Infraorbitals 4 and 5 dorso-posteriorly placed and small, and dermosphenotic larger. One large, squarish suborbital. Small teeth restricted to anterior part of jaws, close to the symphyses between premaxillae and anterior dentalosplenials. Maxilla toothless. Mobile premaxilla. One supramaxilla present. Lower jaw-quadrate articulation placed below the posterior half of orbit. Wedge-shaped operculum. Plate-like, moderately broad preoperculum with gentle antero-ventral curvature, and most ventral region slightly narrower; preopercular canal positioned along the midsection of bone, closer to the posterior margin than to the anterior one, with a few sensory pores. Well-developed triangular interoperculum. Cleithrum with short and characteristically expanded ventral region [*]. Large and expanded clavicle. Markedly narrow caudal peduncle. Abbreviated hemiheterocercal tail. (See Section 3.1.4 for taxonomic comparisons and comments on certain characteristics.)

**Derivatio nominis**: Named after the nearby town Seinstedt (in lower Saxony, Germany).
**Content**: Only one species known, *Seinstedtia parva.*
**Occurrence**: Northern Germany, Upper Triassic (Norian).

#### 3.1.2. *Seinstedtia parva* gen. et sp. nov.

Zoobank registration:
https://www.zoobank.org/urn:lsid:zoobank.org:pub:AB267D05-9918-4E03-876F-FE80F01C93A1 (accessed on 14 October 2022).
**Synonyms**:
1996 "*Semionotus*"-artige Actinopterygier ([13]: p. 147).
2014 *Semionotus*-like actinopterygian ([16]: figure 8a = specimen MLU Sei.2010.76).
2021 cf. *Semionotus* sp. ([30]: plate 1, figure 1 = specimen MLU Sei.2010.76).
**Diagnosis**: See generic diagnosis, plus a few additional specific features: small fish, ca. 50 mm total length, smooth ganoine layer on bones and scales; the latter with straight posterior margin.
**Holotype**: MLU Sei.2010.76 (part; Figure 1A, Figure 2B, and Figure 4) and MLU Sei.2010.77 (counterpart); nearly complete specimen.

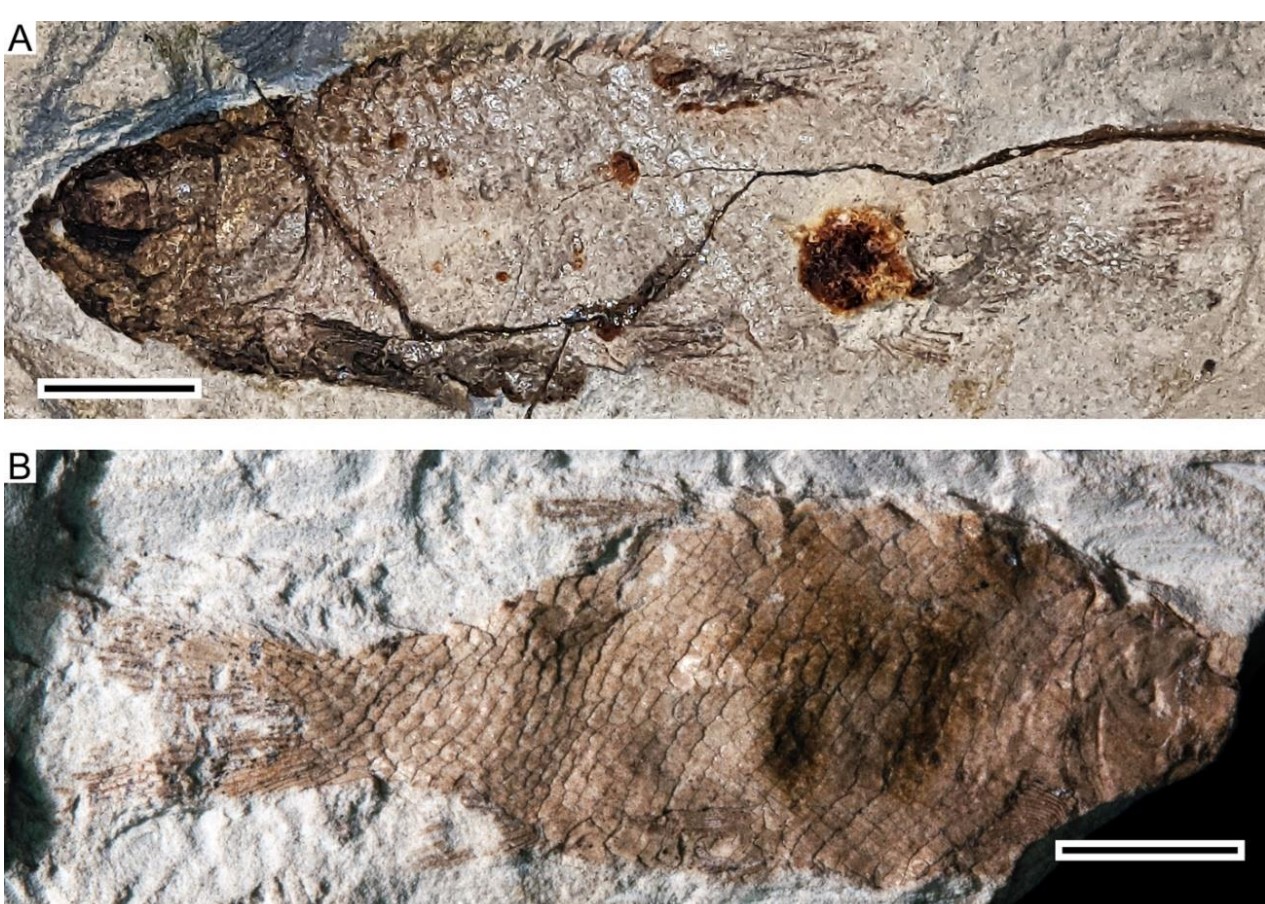

**Figure 1.** *Seinstedtia parva* gen. et sp. nov., from horizon MFL, Section F1, Fuchsberg Quarry near Seinstedt, Lower Saxony, upper Norian, Upper Triassic. (**A**) holotype MLU Sei.2010.76 in lateral view; (**B**) paratype MLU Sei.2010.36 in lateral view. Scale bars = 5 mm. Photograph, courtesy of John Chorn.

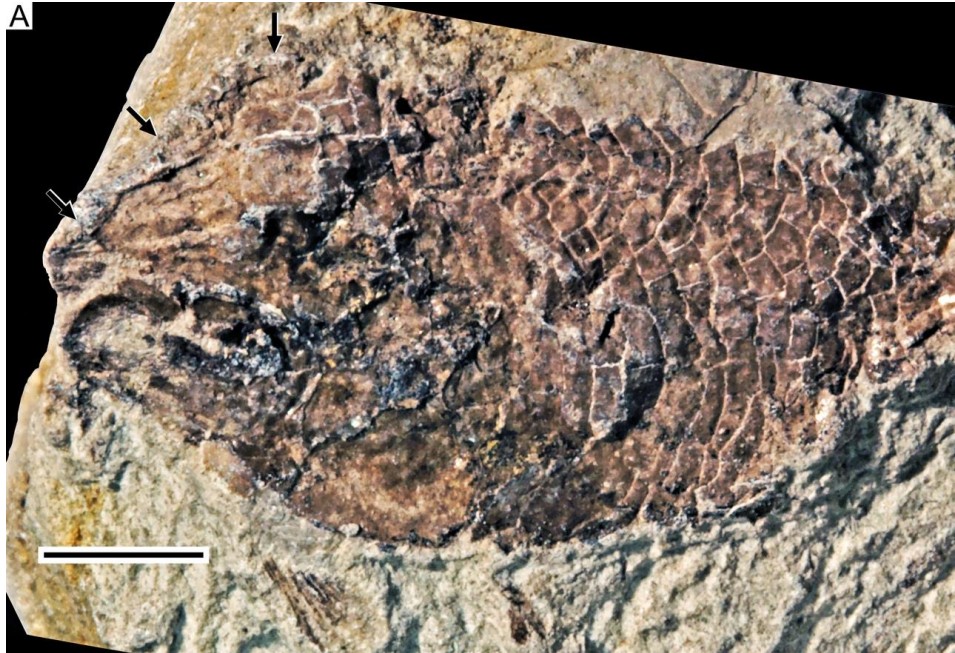

**Figure 2.** *Cont.*

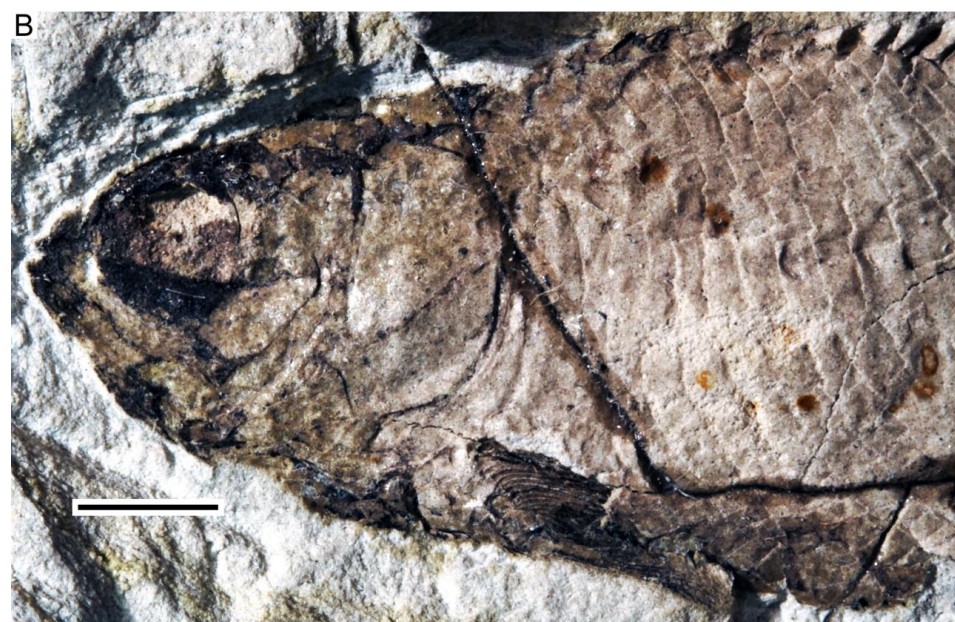

**Figure 2.** *Seinstedtia parva* gen. et sp. nov., from horizon MFL, Section F1, Fuchsberg Quarry near Seinstedt, Lower Saxony, upper Norian, Upper Triassic. (**A**) paratype MLU Sei.2010.83 in dorsolateral view; the skull roof, indicated by arrows, is ventro-laterally displaced; (**B**) head and anterior part of body of holotype MLU Sei.2010.76. Scale bars = 3 mm. Photograph, courtesy of John Chorn.

**Paratypes**: MLU Sei.2010.36, fish missing anterior part of head (Figure 1B and Figure 5); MLU Sei.2010.8080 (part) and MLU Sei.2010.8081 (counterpart), body without head; and MLU Sei.2010.82 (counterpart) and MLU Sei.2010.83 (part; Figures 2A and 3), head and anterior trunk, including squamation.

**Derivatio nominis**: The species name "parva" is feminine, and it means "small" in Latin.

**Provenience**: Horizon MFL, Section F1, Fuchsberg Quarry near Seinstedt, Lower Saxony, Germany, uppermost Norian, Upper Triassic ([16]: Figure 2).

3.1.3. Description

The specimens are preserved in lateral view, except for one specimen (MLU Sei.2010.83) showing the dorsal surface of the skull. Specimen MLU Sei.2010.76 (Figure 1A) is the most completely preserved, and it was figured earlier by Barth et al. ([16]: figure 8a) and Hauschke et al. ([30]: plate 1, figure 1). The holotype and additional specimens show in addition to the lateral side of some cranial bones the imprint of their outer side, making study difficult. The paratype MLU Sei.2010.83 lies on its side in the rock, on bedding plane, and the skull roof is partly displaced over the lower part of the head (Figure 2A), making the description of this area difficult. The counterpart (MLU Sei.2010.82) shows only scales and fin rays. The preservation of bones and their ossification reveals that the specimens were small adults. The preservation of scales in situ does not allow the observation of the vertebral column and its associated elements, pelvic plates, and basipterygia, or dorsal, and anal pterygiophores.

*Seinstedtia parva* is a miniature fish, reaching a total length of 46 mm in specimen MLU Sei.2010.76; however, we estimate the total length at ca. 50 mm, because the distal tips of the caudal rays are incompletely preserved. The head is longer (1.14 times) than deep, and its anterior profile curves gently, giving the anterior part of the head an almost smooth, rounded contour (Figure 1A). The head length is 4.8 times less than total length. The snout is very short, about 10 times smaller than head length, and the orbit is large, ca. 3 times smaller than head length. The maximal body depth of 12.4 mm, and the total length is ca. 3.7 times this depth.

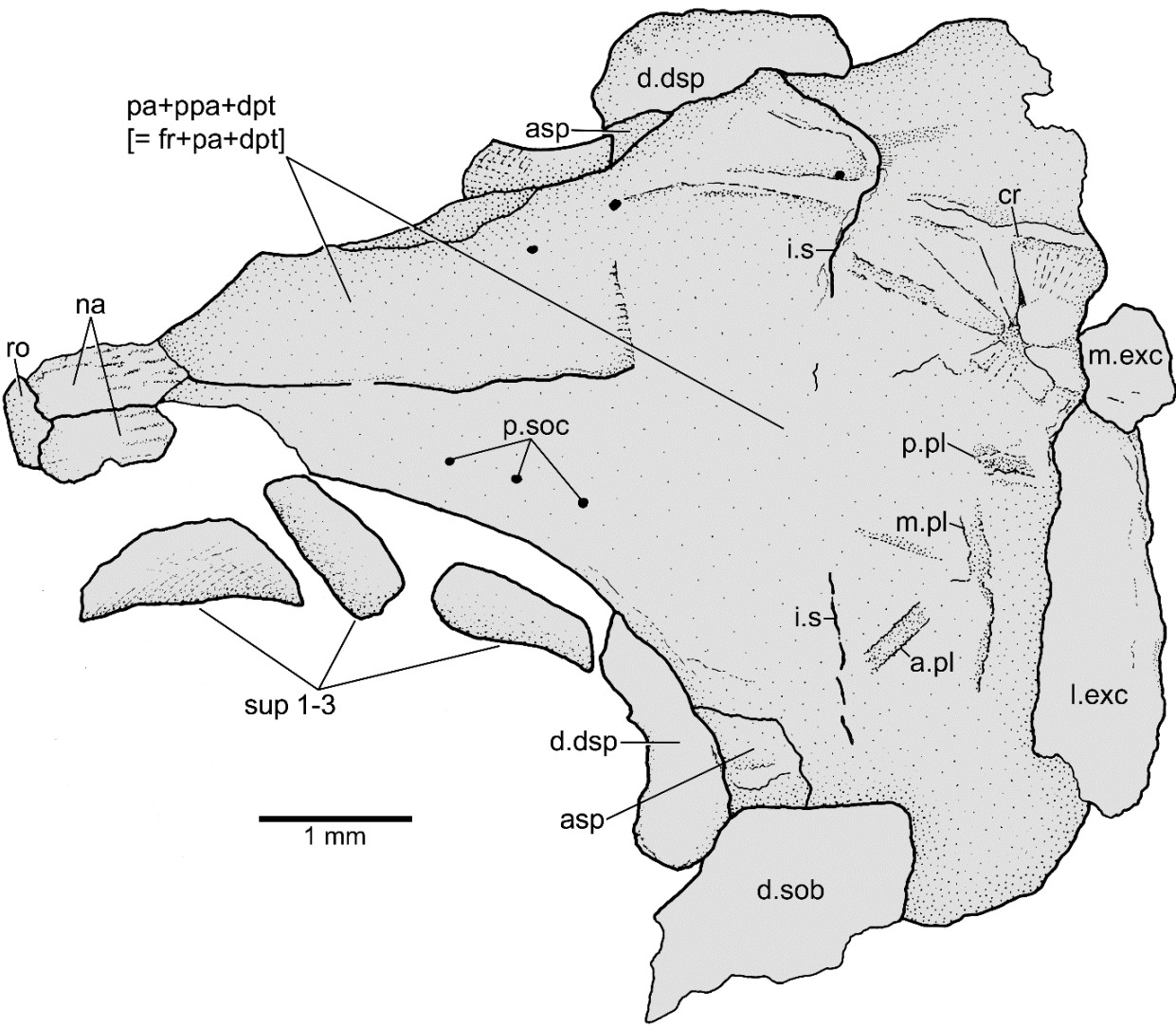

**Figure 3.** *Seinstedtia parva* gen. et sp. nov. Skull roof of paratype MLU Sei.2010.83 in dorso-lateral view. Abbreviations: a.pl, anterior pitline; asp, autosphenotic; cr, crest; d.dsp, displaced dermosphenotic; d.sob, displaced suborbital; i.s, incomplete suture; l.exc, lateral extrascapula; m.exc, median extrascapula; m.pl, middle pitline; na, nasal bone; pa + ppa + dpt [=fr + pa + dpt], parietal + postparietal + dermopterotic [=frontal + parietal + dermopterotic]; p.pl, posterior pitline; ro, rostral; sup1–3, supraorbitals 1–3.

**Skull roof bones and braincase**: All superficial bones are covered with a smooth layer of ganoine, apparently without ornamentation, except for the nasals and posterior tip of the maxilla, which are covered with elongate narrow ridges, and the dentary or dentalosplenial, which appears to have some minuscule, round tubercles close to the symphysis. Under magnification, numerous tiny tubercles are clearly observed in these bones.

An incompletely preserved small rostral bone is placed at the antero-ventral tip of the cranium, articulating posteriorly with the nasal bones (Figure 3) and ventrally with the premaxillae. Both nasals (Figures 2–4) are the main elements forming the gently curved anterior profile of the head. The nasals are rectangular in shape and articulate with each other medially. Due to incomplete preservation, the position of the nostrils is unknown. Posteriorly, the nasal has an oblique articulation with the parietal [=frontal] region, which is the largest region of the skull roof. The parietal region (Figure 3) is about three times the length of the region occupied by the postparietal [=parietal]. The anterior narrow part of the parietal (anterior to the postero-dorsal corner of the orbit) occupies about 60% of its length. The width of the parietal region expands laterad, reaching its caudal width

immediately behind the autosphenotic. The dorsal aspect of the skull roof of specimen MLU.Sei 2010.83 (Figures 2A and 3) reveals a tendency of fusion between bones, with an incomplete smooth suture at the orbital region of the parietals and incomplete sutures between parietals and dermopterotics and parietals and postparietals. It is possible that all bones were fused in older individuals. At the dorso-posterior corner of the parietal region, another bone is observed on the left side; it is interpreted here as the autosphenotic. The postparietal and dermopterotic regions could be traced due to the presence of incomplete sutures and grooves that are interpreted as the anterior and middle pitlines because of their position.

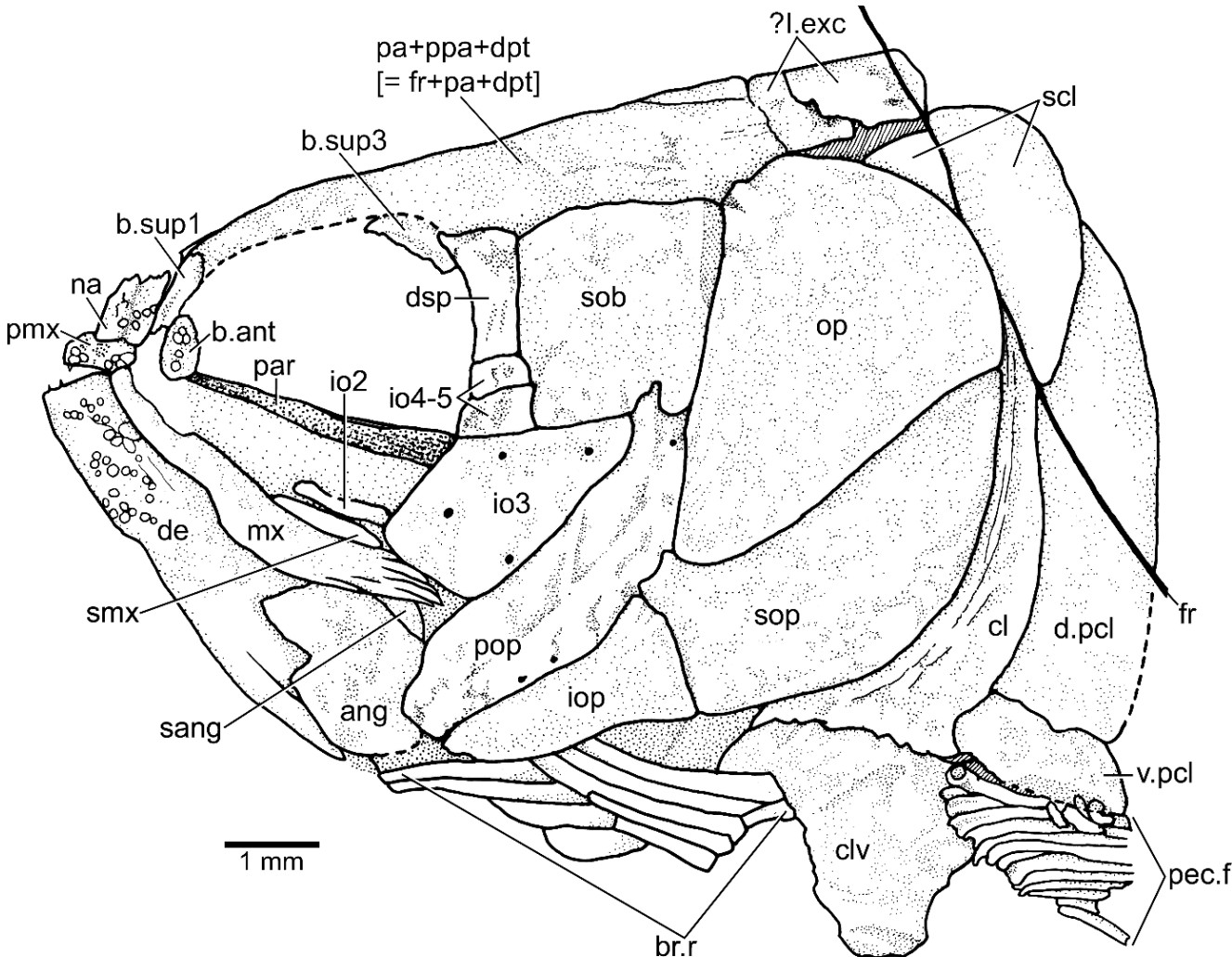

**Figure 4.** *Seinstedtia parva* gen. et sp. nov. Head and pectoral girdle of holotype MLU Sei.2010.76 in lateral view. Abbreviations: ang, angular; b.ant, broken antorbital bone?; br.r, branchiostegal rays; b.sup1, 3, supraorbital bones 1, 3; cl, cleithrum; clv, clavicle; de, dentalosplenial (=dentary); d.pcl, dorsal postcleithrum; dsp, dermosphenotic; fr, fracture; io 2–5, infraorbitals 2–5; iop, interoperculum; ?l.exc, ?lateral extrascapular; mx, maxilla; na, nasal bone; op, operculum; pa + ppa + dpt [=fr + pa + dpt], parietal + postparietal + dermopterotic [=frontal + parietal + dermopterotic]; par, parasphenoid; pec.f, pectoral fin rays; pmx, premaxilla; pop, preoperculum; sang, surangular; scl, supracleithrum; smx, supramaxilla; sob, suborbital; sop, suboperculum; v.pcl, ventral postcleithrum.

The large dermopterotic region (Figure 3) expands ventro-laterally, giving the postero-lateral sides of the skull roof a smooth aspect. The dermopterotic region narrows anteriad where it meets the autosphenotic and dermosphenotic.

The lateral extrascapulae (Figure 3) are ca. four times wider than long and somewhat rectangular shaped. The median extrascapula is as long as wide so that it extends caudad beyond the lateral extrascapulae in specimen MLU Sei.2010.83.

A section of the parasphenoid (Figures 2B and 4) is observed in the orbit. There are no teeth associated with the ventral margin of the parasphenoid, nor with the entopterygoid region. The anterior region of the parasphenoid and vomer is not preserved.

**Circumorbital bones**: The circumorbital ring (Figures 2A, 3 and 4) is incompletely preserved, because the antorbital is partially destroyed and the most anterior infraorbital or infraorbital 1 is missing in the available specimens due to poor preservation. At least three supraorbitals, infraorbitals 2–5, and the dermosphenotic are preserved. The nasal bone is not involved in the orbital margin, but the series is unusual in that the elongated supraorbital 1 forms part of the anterior margin of the circumorbital ring. All circumorbital bones have smooth margins.

Three narrow supraorbitals (Figures 2A and 3) lie in front of the dermosphenotic above and in front of the orbit. They are rectangular in shape. The first one is partially placed lateral to the nasal bone, whereas supraorbitals 2 and 3 are placed lateral to the margin of the parietal region. The most posterior one (supraorbital 3) contacts the dermosphenotic (Figure 4). Two posterior infraorbitals border the orbit posteriorly; the dorsal one or infraorbital 5 is rectangular in shape and slightly smaller than the ventral one or infraorbital 4. Infraorbital 3 (=jugal, after Jollie [31]; figures 2B and 4), at the postero-ventral corner of the orbit, is a large, somewhat rhomboidal bone that occupies the space between orbit and preoperculum; its dorsal border corresponds to the width of posterior infraorbital 4 plus half the width of the suborbital. A poorly preserved narrow bone articulates with the antero-ventral margin of infraorbital 3 and is interpreted here as infraorbital 2 (Figure 4).

One large, squarish suborbital bone (Figures 2B and 4), a little deeper than the two posterior infraorbitals, occupies the space between the posterior expansion of infraorbital 3, posterior margins of infraorbitals 4 and 5, dermosphenotic, and operculum. Accessory suborbitals are absent.

**Upper jaw**: The upper jaw (Figures 2B and 4) is poorly preserved in the available material. It is formed by a mobile premaxilla, maxilla, and supramaxilla. The small premaxilla carries small teeth. The maxilla has a markedly curved and elongate anterior articular region and a slightly curved and short, elongate blade. No teeth have been observed in the oral margin of the blade. The posterior tip of the maxilla is covered by irregular projections separated by narrow grooves that extend close to the anterior margin of the preoperculum. On the middle region of the dorsal margin of the blade lies a narrow supramaxilla. Because of its position and size, it looks like only one supramaxilla was present.

**Lower jaw**: The lower jaw projects slightly in front of the premaxilla, giving the anterior profile of the head a curious aspect. Its complete size is unknown because its dorsal part is covered by the maxilla. As preservation permits, it is possible to observe that the jaw is composed laterally by the dentary or dentalosplenial, angular, and surangular (Figures 2B and 4). The dentalosplenial has small teeth near the symphysis that are opposed to the small teeth of the premaxilla. The dentalosplenial deepens posteriad until it meets the angular in a zigzag suture. Its ventro-posterior region extends close to the posterior tip of the angular. A portion of the surangular is observed in one specimen (Figure 4) at the dorsal region of the angular. It is unknown whether a high coronoid process was present and if a "leptolepid" notch is on the ascendent margin of the dentalosplenial.

**Opercular bones**: The opercular series includes the preoperculum, operculum, suboperculum, and interoperculum. The preoperculum, a gently antero-ventrad arched bone, is placed posterior to infraorbital 3 and lower jaw. The dorsal part of the bone is partially covered by the large suborbital, so that the complete size of the preoperculum is unknown. The preoperculum (Figures 2B and 4) is a plate-like, relatively broad bone, with a gentle ventro-anterior curvature and its most ventral tip slightly narrower. The preopercular

sensory canal is positioned at the mid-region of the bone, and at least three small round pores are preserved in MLU.Sei 2010.76 close to the posterior margin of the bone.

The operculum (Figures 2B and 4) is wedge-shaped, pointing antero-ventrad so that the ventral border with the suboperculum is oblique. The dorsal margin of the operculum is rounded. The width of the operculum is about 2/3 of its length, and the bone is slightly larger than the suboperculum.

The suboperculum (Figures 2B and 4) is longer than deep, with a short and broad antero-dorsal process. The triangular-shaped interoperculum (Figures 2B and 4) is large and sutures posteriorly with the whole anterior margin of the suboperculum, extending anteriad, nearly to the anterior tip of the preoperculum. The suboperculum is surrounded posteriorly by the cleithrum and the operculum by the supracleithrum.

**Hyoid arch, branchiostegal rays, gular plate, and urohyal**: The hyoid arch is not preserved, but nine branchiostegals rays are incompletely preserved in the holotype (Figures 2B and 4). No gular plate or urohyal have been observed in the studied material, and it is unknown if these bones were present or not.

**Cephalic lateral line system**: Few traces of the lateral line system are preserved. A few small, rounded pores of the supraorbital canal are visible in the anterior orbital portion of the parietal region (Figure 3), closer to the orbital margin than to the incomplete interparietal suture. The supraorbital canal apparently does not reach the postparietal region because its parietal branch seems to be absent. Remnants of the anterior and middle pit-lines are observed (Figure 3); no traces of the middle pitline are observed on the dermopterotic portions of the skull roof. Part of the trajectory of the otic canal is observed on the anterior half of the dermopterotic.

The infraorbital canal (covered by thin bone) is positioned at the mid-region of the dorso-posterior infraorbitals 4–5 and the dermosphenotic and continues in the dermopterotic as the otic canal. The supraorbital and infraorbital canals do not join in the dermosphenotic. The infraorbital canal branches in the infraorbital 3, and a few small round pores (Figures 2B and 4) are observed near its anterior and posterior margins.

The trajectory of the preopercular canal is covered by bone (Figures 2B and 4); the canal runs closer to the posterior margin of the bone than to the anterior one and branches in a few short tubules (ca. 3) that open to the surface throughout small, rounded pores.

**Shoulder girdle and pectoral fin**: The posttemporal, the bone linking the skull roof and the pectoral girdle, is described in this section. The poorly preserved posttemporal seems to be a large and broad bone without evident processes for articulation with the skull. The supracleithrum and cleithrum (Figures 2B and 4) follow the contour of the opercular region; the supracleithrum encompasses the operculum, whereas the cleithrum is elongated and markedly short ventrally. The cleithrum reaches near the ventral border of the suboperculum where it articulates with the clavicle. There are a few serrated ridges on the cleithrum running parallel to the posterior margin of the suboperculum. The remarkably short lower part of the cleithrum is widened and has a slightly serrated articulation with the clavicle. The clavicle is a massive, laterally expanded bone. One deep, broad, scale-like postcleithrum is placed posterior to the cleithrum; a second small and narrow ventral postcleithrum is placed ventral to the main dorsal postcleithrum. The width of the large postcleithrum is double of that of the following scales; it also exceeds the width of the exposed part of the cleithrum.

The pectoral fins (Figure 1A) are placed closer to the ventral margin of the body than to the mid flank. The pectoral fin, although incomplete, seems to be small and composed of 13–16 fin rays. The fin rays are segmented and distally branched, at least the first rays. The first ray (Figure 4), which is partially exposed in the holotype, has an expanded proximal region. The propterygium is not preserved, a fact that can be interpreted as the propterygium not being fused with the proximal region of the ray. Because of the incomplete preservation of this region, it is unclear whether basal and/or fringing fulcra were present or not.

**Pelvic girdle and fin**: Due to the preservation of the scales in situ, the basipterygium or pelvic plate is not exposed in the available specimens. Each pelvic fin is small, with eight or nine thin, delicate rays that are segmented distally, and apparently some are branched distally. Specimen MLU Sei.2010.77 has small and fine fringing fulcra associated with the pelvic fins.

**Dorsal and anal fins**: Dorsal and anal fins (Figure 1A,B) are incompletely preserved; they may have a triangular shape. The dorsal fin lies in the middle of the body, with its origin slightly behind the origin of the pelvic fins. The dorsal fin has 14 or 15 narrow fin rays, and the posterior ones are spaced apart from each other. A few fine, elongate, fringing fulcra are associated with the leading margin of the fin. Segmentation and branching occur in the distal parts of the fin rays. A small scute precedes the dorsal fin.

The anal fin has 12 fin rays, which are segmented and branched distally.

**Caudal fin**: The caudal fin (Figures 1B and 5A,B) is preserved in MLU Sei.2010.36 and 2010.81. In both specimens, the posterior tips of the rays are missing or are damaged. The dorsal body lobe extends shortly (ca. four scales deep) and ends posteriorly in a sharp tip, whereas the ventral body lobe is short and gently rounded. The fin is hemiheterocercal, with the posterior margin furcated. There are seven to nine dorsal basal fulcra that are elongate and lanceolate. It is unclear whether they are unpaired or paired basal fulcra because they are preserved in lateral view in both specimens. There are at least 18 principal caudal rays; the number is imprecise because it is not possible to identify with confidence the last principal ray. Additionally, the presence of ventral procurrent rays cannot be determined with confidence, and nor can the total number of ventral basal fulcra, which seems to be three or more. A series of small, elongate fringing fulcra are observed in the dorsal and ventral lobes of the fin. A few additional fulcra are observed in the ventral lobe. The ray that is interpreted here as the possible last principal ray shows in the posterior segments a fulcrum-like projection (Figure 5B) that may alternate with the fringing fulcra.

Dorsal and ventral scutes (Figure 5A) precede the fin. The scutes are incompletely preserved, but they are elongate and broad. It is unclear whether the two small oval scales at the distal tip of the dorsal body lobe can be interpreted as urodermals (Figure 5A).

**Body squamation**: The body is covered with 31 or 32 vertical rows of rhombic ganoid scales. The free field of the scales is covered with smooth ganoine. The posterior margin of the scales is straight, and the ventro-posterior corner forms a pointed spine in some scales. The posterior margin of the lateral line scales moves forward at the invagination for the exit of the lateral line. The lateral line scales are deeper than long (being the free posterior field more than twice as deep as long), as are the scales of the two horizontal lines below the lateral line scales. There are a maximum of 14 scales in one vertical row, four to five scales above the lateral line and eight to nine below it. There are eight to nine spike-like dorsal ridge scales in front of the dorsal fin of the holotype (Figure 1A). However, that is not the case in the other specimens, as seen clearly in specimen MLU Sei 2010.83, where the scales are exposed in dorsal view (Figure 2A); the spine-like appearance is a preservation artifact as is shown in specimen MLU Sei.2010.36 (Figure 1B) with similar 'elements' above the basal fulcra. The pterygial formula is:

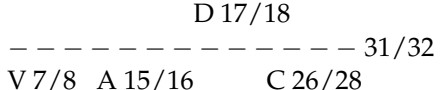

$$\begin{array}{c} \text{D } 17/18 \\ {-\,-\,-\,-\,-\,-\,-\,-\,-\,-\,-\,-\,-}\;\; 31/32 \\ \text{V } 7/8 \quad \text{A } 15/16 \qquad \text{C } 26/28 \end{array}$$

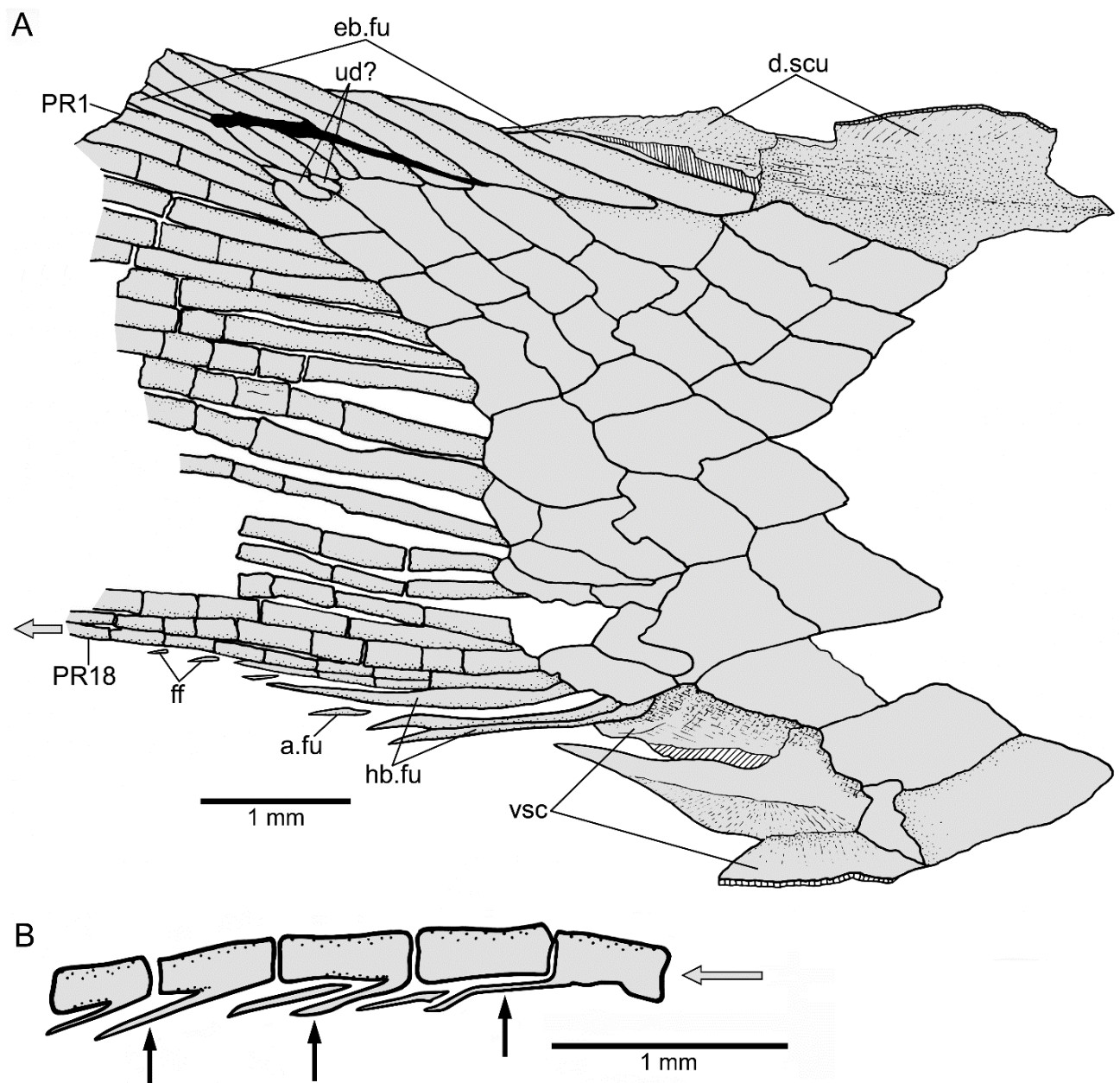

**Figure 5.** *Seinstedtia parva* gen. et sp. nov.; paratype MLU Sei.2010.36 in lateral view. (**A**) caudal fin and region in lateral view; (**B**) detail of a posterior segment of the last principal ray as indicated by gray arrows in (**A**,**B**). Black arrows point to projections of the caudal segments that look like fringing fulcra. Abbreviations: a.fu, additional fulcrum; d.scu, dorsal scute (broken); eb.fu, epaxial basal fulcra; ff, fringing fulcra; hb.fu, hypaxial basal fulcra; PR1–18, principal rays 1–18; ud?, urodermals?; vsc, ventral scute (broken). Black area indicates a fracture.

### 3.1.4. Comparisons and Taxonomic Assignment

　　The small actinopterygian fishes from Seinstedt are difficult to place systematically because of the combination of morphological characteristics and the lack of information of some characteristics due to incomplete preservation. Previously, the fishes were interpreted as semionotiform-like or possibly *Semionotus*, mainly based on the supposedly "spiny" median ridge scales in the predorsal region of the trunk, but as explained above, these are artifacts of preservation, not real dorsal median processes of the median predorsal scales (Figures 1B and 2A). However, when this new and more detailed morphological description of the fishes is compared with semionotiforms (e.g., in [32–38]), we do not

find support for this taxonomic assignment. The combination of characteristics (especially cranial configuration and features) of the studied specimens does not place them close to any semionotiform taxon. Despite earlier assignments, *Seinstedtia parva* gen. et sp. nov. does not belong to Semionotiformes and cannot even be placed within the Ginglymodi. Consequently, a broader survey of Triassic fishes with characteristically shaped skull roof and preoperculum, mobile premaxillae, supramaxillary bone(s) present, and other features was conducted, and their morphological characteristics compared. From the results of these comparisons, the search was restricted to a few groups: parasemionotids, such as *Phaidrosoma lunzensis* Griffith, [39] from Austria; a small fish from Switzerland (*Prosantichthys* Arratia and Herzog, [40]); the ichthyokentemid *Elpistoichthys* Griffith [39]; and a variety of Triassic pholidophorids from Italy (e.g., [41–47]) and Austria [39,45].

A comparison with European Late Triassic Parasemionotiformes does not support *Seinstedtia parva* gen. et sp. nov. as one of its members because of major differences in the configuration of the skull roof, absence of supraorbitals, configuration and number of the dorso-posterior infraorbitals and suborbital(s), and structure of the preoperculum and the cleithrum, among others (compare Figure 4 with figures 22 and 23 in [39] and figures 3 and 4 in [40]).

The Late Triassic *Elpetoichthys pectinatus*, interpreted as an ichthyokentemid by Griffith [39], shares with *Seinstedtia parva* gen. et sp. nov. the smooth, gently curved antero-dorsal profile of the head that includes the anterior tip of the parietal bone, the nasal, a small rostral, and a small premaxilla; the antero-dorsal margin of the circumorbital ring formed by an enlarged first supraorbital; the arrangement of the postero-dorsal infraorbitals 4 and 5 that are smaller than the larger dermosphenotic; one big suborbital bone; one supramaxilla; a skull roof with partially fused bones according to the holotype specimen (NHMW 2007z0170/0227; contra Griffith [39]: figure 25 with all skull bones independent); and three extrascapulae instead of two (according to specimen NHMW 2007z0170/0227; contra Griffith [39]: figure 25 showing only two broad extrascapulae). Despite these shared characteristics, the two taxa present some major differences: a skull roof with a narrower parietal region followed by an expanded postorbital region in *Elpistoichthys* that is not as expanded as in *Seinstedtia parva* gen. et sp. nov.; nasal bones narrowly sutured to each other, whereas they are sutured for most of their length in *Seinstedtia parva* gen. et sp. nov.; one enlarged supramaxilla in *Elpistoichthys,* which is narrower and smaller in *Seinstedtia parva* gen. et sp. nov.; and shape and size of the preopercular bone, which is broader and larger in *Elpichtoichthys* and with many elongate sensory tubules and pores, in contrast to *Seinstedtia* where the bone is narrower and with few (ca. 3) preopercular pores. The lower region of the cleithrum is longer in *Elpichtoichthys*, whereas it is characteristically shorter and broader (Figure 4) in *Seinstedtia parva* gen. et sp. nov.

*Seinstedtia* agrees with many characteristics of Pholidophoridae (node C1 in Arratia [46]: figure 95), such as skull roof bones fused, skull roof in orbital region narrower than in postorbital region, and a wedge-shaped operculum. Internal vertebral characteristics and fin characteristics cannot be checked because they are not preserved. Single pholidophorid genera have additional characteristics in common with *Seinstedtia*. *Pholidophoretes salvus* from the Carnian of Polzberg bei Lunz, Austria, shares with *Seinstedtia parva* gen. et sp. nov. some important characteristics: the ganoine ornaments of dermal cranial bones are almost nonexistent; a characteristic pholidophoriform-shaped cranial roof with narrow orbital region and largely expanded postorbital region; fusion between skull roof bones; nasal bones well developed and with extensive medial contact; three postero-dorsal infraorbitals; infraorbital 3 the largest and extending below the single and enlarged suborbital; and rhomboidal shaped ganoid scales, with a smooth surface and posterior margin. Despite these important similarities, there are major differences between the taxa: the straighter snout with the nasals and rostral almost horizontally oriented in *Pholidophoretes*, in contrast to the gently rounded anterior profile of the head of *Seinstedtia parva* gen. et sp. nov.; maxilla with a notched distal end in *Pholidophoretes* ([46]: figures 92 and 94), in contrast to a slightly elongate posterior margin (Figure 4) in *Seinstedtia parva* gen. et sp. nov.; presence

of two enlarged triangular extrascapulae in *Pholidophoretes* ([46]: figure 92), in contrast to three narrow extrascapulae (Figure 3) in *Seinstedtia parva* gen. et sp. nov.; suboperculum larger than operculum ([46]: figure 92) in *Pholidophoretes* versus operculum larger than suboperculum in *Seinstedtia parva* gen. et sp. nov.; preoperculum with numerous sensory tubules that end in a rounded pore along the bone ([46]: figures 92 and 93) in *Pholidophoretes* versus a preoperculum with a few (ca. three) pores in *Seinstedtia parva* gen. et sp. nov.; and dorsal fin positioned opposite to anal fin in *Pholidophoretes*, in contrast to a dorsal fin placed about the mid-length of the body, almost anterior to the origin of the pelvic fins in *Seinstedtia parva* gen. et sp. nov.

Pholidophorids, as well as ichthyokentemids, are currently interpreted as teleosteomorphs in available phylogenetic hypotheses (e.g., [46,47]), and *Seinstedtia parva* gen. et sp. nov. shares with them, as well as with other stem teleosts, the presence of a mobile premaxilla, the shape of the skull roof with the orbital region narrower than the postorbital one, reduction in number of infraorbital bones to five, and one or two supramaxillae. According to the available information based on the description above, the fossil studied here differs from other stem teleosts or teleosteomorphs in numerous characteristics that justify its inclusion in a new genus and species, *Seinstedtia parva*.

### 3.1.5. Miniaturization and Small Body-Sized Fishes

Actinopterygian fishes are characterized by different body sizes, even in one family the range may run the gamut from miniature species with less than 2.6 cm standard length (SL) to large species over a few meters. Miniaturization (reaching sexual maturity at 20 mm SL or less and not growing longer than 26 mm SL) is a well-known phenomenon that characterizes numerous, extant teleostean families and includes fishes that may exhibit paedomorphism. These conditions may pose a problem when studying the fish fossil record, because fossilization of tiny, delicate fishes is unlikely, as reflected in the rarity of reported larval stages. Additionally, determining the time of sexual maturity in fossil fish is a difficult goal—even estimations of length may be problematic because often, the total length, not the standard length, is given for fishes having a heterocercal or hemiheterocercal caudal fin. A number of small fossil fishes have been recovered in Triassic strata. Although technically none may be "miniature" fishes, greater analyses will be important in understanding the evolution of some groups.

The Late Triassic fish described herein, *Seinstedtia parva* gen. et sp., reached less than 50 mm total length, and because of its size, it can be considered a small body-sized species. This miniature fish, together with other small body-sized taxa of ca. 70 mm total length (*Parapholidophorus nybelini* and *Pholidoctenus serianus* from Italy), inhabited aquatic ecosystems of Europe during the Norian. According to the available information, most known Triassic teleosteomorphs were small, ranging from ca. 40 to 85 mm total length, except for *Knerichthys bronni* (borders of Austria and Slovenia; Carnian) and *Pholidorhynchodon malzani* (Italy; Norian) that were ca. 140–150 mm total length. Among the smallest teleosteomorphs, between 40 and 50 mm total length, are *Prohalecites porroi* (Italy), *Marcopoloichthys andreetti* (Italy), *M. faccii* (Italy), and *M. furreri* (Switzerland [48–50]).

This Triassic small body-sized condition was not unique to teleosteomorphs because small species of non-teleosteomorph actinopterygians are known from the Triassic of Europe, for instance, the neopterygians *Habroichthys minimus* (ca. 32 mm total length, which probably reached sexual maturity at 26 mm standard length and is a candidate for a miniature fish), *Peltopleurus notocephalus* (ca. 45 mm total length), and *Peltoperleidus macrodontus* (ca. 50 mm total length [51]), and the parasemionotiform *Prosantichthys buergini* (ca. 60 mm total length [40]). A variety of small body-sized neopterygians are also known from the Triassic of China, for instance, the neopterygian incertae sedis, *Frodoichthys luopingensis* and *Gimlichthys dawaziensis*, (about 40 mm total length and a candidate for a miniature fish [52]); the thoracopteroid *Wushaichthys exquisitus* (ca. 57 mm total length [53]); and the louwoichthyiform *Peltoperleidus asiaticus* (ca. 46 mm total length [54]).

### 3.1.6. Diversity and Paleoecology

Triassic teleosteomorphs are known from China and Europe, with the oldest ones *Prohalecites* from the Ladinian of Italy and a pholidophoriform—*Malingichthys*—from the late Ladinian of China [55]. Among teleosteomorphs or stem teleosts, *Prohalecites* and the pholidophorids *Annaichthys*, *Eopholidophorus*, *Knerichthys*, *Lombardichthys*, *Parapholidophorus*, *Pholidorhynchodon*, and *Zambellichthys* are only known from Italy [46–48], whereas the pholidophorids *Pholidophorus* and *Pholidophoretes* and the ichthyokentemid *Elpetoichthys* are known from Austria [39,46]. *Pholidoctenus,* which is known from numerous and well-preserved specimens from Italy, has been reported from one specimen from the Middle Triassic of Germany [56]. According to current information, marcopoloichthyids are the teleosteomorphs with the broader distribution, being known from China, Italy, and Switzerland, with higher species diversity in Italy [49,50]. Among these teleosteomorphs, the most diverse are the pholidophorids. According to our best information, *Seinstedtia parva* gen. et sp. nov. would be the second record of a Triassic teleosteomorph for Germany.

All Triassic teleosteomorphs mentioned above are known from marine deposits in China and Europe. Arratia [46] described (1) *Pholidophorus gervasuttii*, *Zambellichthys bergamensis*, *Annaichthys pontegiurinensis*, *Pholidorhynchodon malzannii*, *Parapholidophorus nybelini* and *P. caffii*, and *Pholidoctenus serianus* from marine Norian deposits in Lombardy, Italy; (2) *Knerichthys bronni* (Undine) from Upper Triassic marine deposits in the southern Italian Alps (see also [36]); and (3) *Pholidophorus latiusculus* and *Pholidophoretes salvus* from Upper Triassic marine deposits of the Austrian Alps. One skull roof of *Pholidoctenus* sp. from the lower Muschelkalk (Anisian, Middle Triassic) of Rüdersdorf near Berlin was illustrated by Schultze and Kriwet ([56]: figure 11). This occurrence indicates an earlier connection between alpine and Muschelkalk Triassic, which is also documented by co-occurrence of *Saurichthys* and *Birgeria* in both regions.

### 3.2. Actinistians

Sarcopterygii Romer, 1955 [57].
Actinistia Cope, 1871 [58].

### 3.2.1. Actinistia Indet

**Synonym**:
2014 coelacanth scale. ([16]: figure 8e). (= MLU Sei.2010.78/79).
**Studied material**: Two scales catalogued as MLU Sei.2010.63 (part) and MLU Sei.2010.64 (counterpart); and MLU Sei.2010.78 (part) and MLU Sei.2010.79 (counterpart).
**Provenience**: Horizon MFL, Section F1, Fuchsberg Quarry near Seinstedt, Lower Saxony, Upper Norian, Upper Triassic.
**Description**: Scale MLU Sei.2010.78 (part; Figure 6) and 79 (counterpart) is complete; it is ovoid with a posterior pointed end of the free field. The scale is 13 mm long and 9 mm deep. The second specimen is represented only by the free field of scale MLU Sei.2010.63 (part; Figure 7) and 64 (counterpart), and this scale is 15 mm deep. The scales are elasmoid of amioid type, which means that the covered field is formed by anterior radiating striae. The free field is covered by elevated, parallel running, elongated ridges typical of actinistian scales. There are 16 ridges on the smaller scale and 26 on the larger scale. The difference in the number of ridges corresponds to the size difference.

### 3.2.2. Comparison and Taxonomic Remarks

The occurrence of ridges on the free field is a common feature on actinistian scales. As far as is known in the Triassic, parallel ridges occur on scales of 25 actinistian species (e.g., [59] for *Diplurus*; see also [60]). The Seinstedt scales with 16 and 26 ridges lie in the middle between scales with few (up to 10) ridges that are widely space, and scales with many (more than 30) ridges that are packed. In the Late Triassic, two of seven species with parallel ridges have comparable numbers of ridges: *Diplurus newarki* (13 ridges [59]) and *Guizhoucoelacanthus guanlingensis* (20 ridges [61]). The scales of *D. newarki*, like those

of *Chinlea sorenseni* [62], are distinct from the Seinstedt scales by a pronounced, elevated median ridge and from *G. guanlingensis* by the presence of tubercles. None of the scales of Triassic coelacanths closely resembles those from Seinstedt. However, one must consider that species within one genus can have different scale ornamentation ([63]: figures 11 and 19) and that the ornamentation may change in different regions of the body ([60]: figure 2).

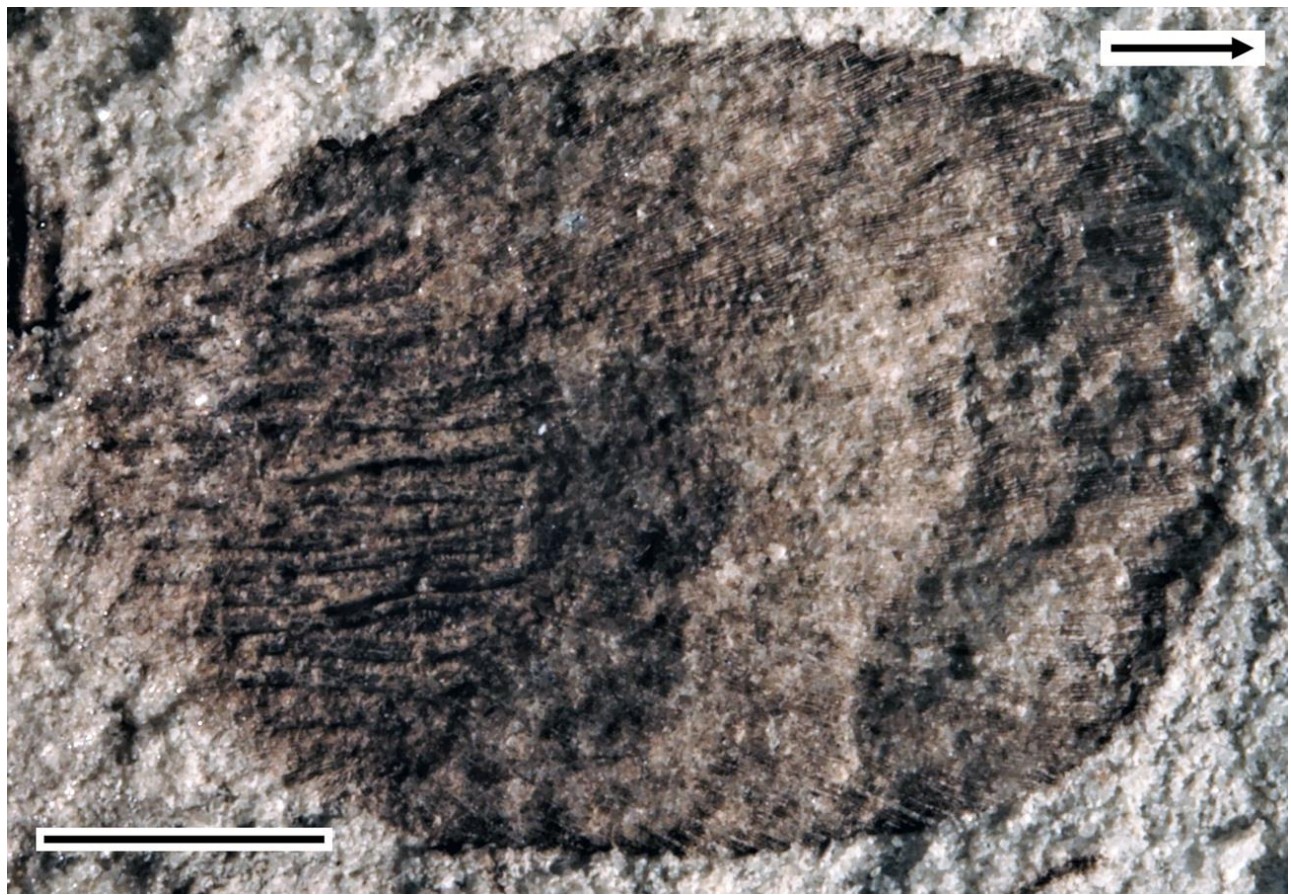

**Figure 6.** Actinistian scale, specimen MLU Sei.2010.78 from horizon MFL at Fuchsberg Quarry near Seinstedt, Lower Saxony, upper Norian, Upper Triassic. Arrow points anteriad. Scale bar = 3 mm. Photograph, courtesy of John Chorn.

### 3.2.3. Comments on Triassic Actinistians

The Triassic is the period with more actinistian genera and species than any other period [64–67]. Just after the P/T extinction, there were 15 marine genera, including 24 species in the Early Triassic; this diversity shows more genera and species than in any other period. This high number is followed by 12 species in the Middle Triassic and eight in the Late Triassic. Interestingly, mostly complete actinistian specimens are recorded during the Triassic; rarely, species are known based on parts of the head and only occasionally scales [68–70], even though Böttcher [70] argued that isolated actinistian remains are more common but scales are too fragile to be preserved and recovered.

### 3.3. Paleoecology

Most Triassic coelacanths are recorded from marine deposits. All Early Triassic genera (15) and species (24) occur in marine deposits. Among them, one species and genus (*Moenkopia*) among the 12 Middle Triassic species is known from freshwater deposits [71] and three species of eight Late Triassic species in freshwater deposits [59,62,72]. Böttcher [70] described and figured many actinistian bones and scales from different horizons of the Lettenkeuper (upper Middle Keuper) representing deposits of different salinity.

Most actinistians occur in marine to brackish deposits, although some are found in deposits of low salinity. Most of the occurrences of coelacanths in freshwater deposits are recorded from North America (*Moenkopia, Chinlea,* and *Diplurus* [59,62]; and *Quayia* [72]) and two undetermined actinistians from the Upper Triassic of Morocco [73] and Germany [74,75]. A rich fish fauna is associated with *Chinlea* in western North America [76], including marine forms such as *Gyrolepis* and *Australosomus,* that indicates connection to a marine environment. The two scales of Seinstedt cannot give a clear indication of a depositional environment. A connection to a marine environment may be likely based on the prevalence of marine actinistians in the Triassic, and Barth et al. [16] interpreted the locality in the Late Triassic as a fluviatile to brackish environment close to the marine basin northeast of Seinstedt.

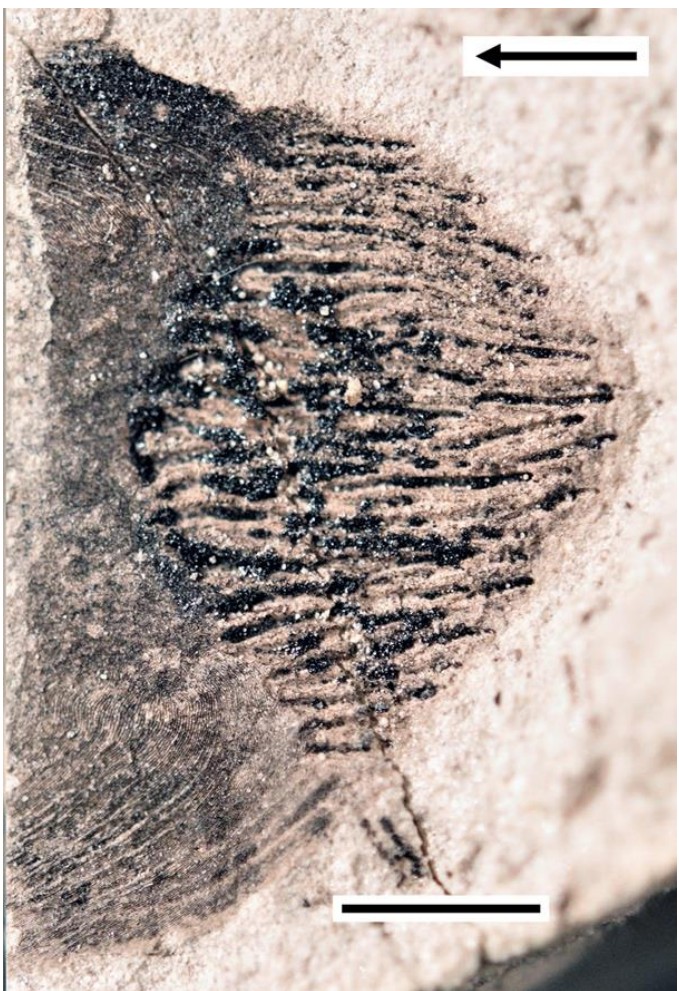

**Figure 7.** Actinistian scale, specimen MLU Sei.2010.63 from horizon MFL at Fuchsberg Quarry near Seinstedt, Lower Saxony, upper Norian, Upper Triassic. Arrow points anteriad. Scale bar = 3 mm. Photograph, courtesy of John Chorn.

## 4. Conclusions

The fish diversity of Fuchsberg Quarry near Seinstedt, lower Saxony, Germany, now includes 12 taxa grouped into two main lineages that occur in two horizons: chondrichthyans (5) and osteichthyans (7). The osteichthyans are represented by actinistian scales and mainly by a variety of actinopterygians, for instance: (1) a palaeoniscimorph (*Gyrolepis albertii*); (2) generalized or primitive neopterygians, including one chondrostean (*Birgeria acuminata*), one colobodontid (*Colobodus* sp.), and one perleidiform (*Serrolepis suevicus*); and (3) the current addition of an advanced neopterygian, a teleosteomorph or stem teleost. The actinistian scales and the advanced neopterygian occur within the Fuchsberg Quarry proper,

whereas chondrichthyans and other actinopterygians were found in a locality higher in the section outside the quarry.

A teleosteomorph, *Seinstedtia parva* gen. et sp. nov., is described based on a few complete and partially preserved specimens that were previously interpreted as semionotiform-like. After a careful study of the specimens, which are not very well preserved, their characteristics were compared with a variety of Triassic neopterygians, and the results of this survey led us to interpret them as a new stem teleostean taxon not assignable to semionotiforms.

The new genus and species are supported by a unique combination of features, for instance, characteristically shaped skull roof, narrow at the orbital region and considerably expanded postorbitally; fusion of skull roof bones; characteristically shaped anterior profile of head; three supraorbital bones with the first one forming most of the antero-dorsal margin of the circumorbital ring; reduction of infraorbital bones to five, three of which are dorso-posterior; infraorbital 3 the largest; movable premaxilla; one supramaxilla; three extrascapular bones; cleithrum with short and greatly expanded antero-ventral region; massive and enlarged clavicle; and ganoid scales with smooth surface and posterior margin. With this combination of characteristics, *Seinstedtia parva* cannot be assigned to any family, and it is interpreted here as incertae sedis within Teleosteomorpha.

*Seinstedtia parva* gen. et sp. nov. is a small fish, less than 50 mm maximum length and is interpreted here as an adult small body-sized fish that, together with other small fishes of less than 70 mm length, such as the pholidophoriforms *Parapholidophorus nybelini* and *Pholidoctenus serianus*, inhabited Europe during the Norian. There were a number of small body-sized neopterygians living in Europe and Asia during this time; its biological significance during the Middle and Late Triassic is a subject that needs further investigation and is currently under study by one of the authors. The fish diversity of Fuchsberg Quarry near Seinstedt with the occurrence of taxa known from marine environments in other places indicates occasional connection to a marine environment during the uppermost Norian.

**Author Contributions:** H.-P.S. and G.A. contributed to the design, study of the fishes and their taxonomy, descriptions and illustrations, and to the general discussion on the taxonomic part and biological interpretations; N.H. and V.W. collected the fishes studied and contributed to the geology, environmental interpretations, and history of Fuchsberg Quarry and fossiliferous content. All authors have read and agreed to the published version of the manuscript.

**Funding:** This research received no external funding.

**Institutional Review Board Statement:** Not applicable.

**Data Availability Statement:** Not applicable.

**Acknowledgments:** Our thanks to Jürgen Kriwet for his invitation to contribute to this special issue on "Evolution and Diversity of Fishes in Deep Time". The authors thank Ursula Goehlich (Vienna); Anna Paganoni (Bergamo); and Heinz Furrer (Zürich) for access to specimens in collections under their care and for providing high-quality photographs. We thank John Chorn, University of Kansas, Biodiversity Institute, who photographed the specimens studied here; Julia Türtscher, University of Vienna, Department of Palaeontology, for her kind assistance formatting the references; and Terry J. Meehan (Lawrence, KS, USA) for revision of the style and grammar of the manuscript.

**Conflicts of Interest:** The authors declare no conflict of interest.

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
