# Peer review of "Osteichthyan Fishes from the uppermost Norian (Triassic) of the Fuchsberg near Seinstedt, Lower Saxony (Germany)"

_diversity, doi:10.3390/d14110901_

Round 1

Author Response

We thank the reviewer for the time and effort spent reviewing the manuscript (ms.). Rev. 1 did not provide a list of comments, but these were included directly in the ms. We note here, as well in comments of other reviewers, that numerous changes in the layout of the ms. were automatically done by the uploading system and/or by saving the submitted manuscript. These changes created a number of problems in the presentation of the ms, most of which were corrected one by one in the revised version, except for the spaces left between the description of some cranial bones and Figure 4 (pages 8-9), the caudal fin and Figure 5 (pages 11-12), and actinistians (Figure 8, pages 17–18) that we are unable to correct and asked the editor that this problem be corrected by their office.  

We considered all editorial comments provided by Rev. 1:

  1.  Page 3. Material and methods. The first sentence was deleted following suggestion of Rev. 1.
  2. Page 3. Genus’ name appeared in normal font; it was changed into italics. ALL genera and species names were double checked along the ms.
  3. Page 3. Diagnosis. Reviewer 1 asked to add differences to comparable taxa. The diagnosis establishes from its beginning that it is based on a unique combination of characters, and unique characters separating the new taxon from other teleosteomorphs are identified by an asterisk. Three characters are uniquely derived and properly identified in the diagnosis. In addition, section 3.1.4 extensively discusses the characters of the new taxon with comparable taxa. A note guiding the reader was added to the end of the Diagnosis.
  4. Page 3. Presentation of Synonyms. Rev. 1 complained about this presentation, that commonly includes the whole name of authors; however, this is not the presentation of literature in this journal, and we are following the Instructions for Authors of the journal. However, we added the word Synonyms and slightly modified the spacing following the format of the journal.
  5. Page 5. The caption of Figure 1 appears at the beginning of the following page. Obviously, this is a problem originated during the uploading, because this lack of continuity was not in the submitted original ms. A similar problem also occurred with Figure 4 (partially). These problems were corrected in the revised ms., but the person responsible of the layout in the journal should check the whole layout for consistency in presentation.
  6. Page 12. The space separating the last line of the caption of Figure 5 was lost, and caption and text looks continuous. This is also a technical problem produced during the uploading/ preparation of the layout, because it was not in the submitted ms. We have fixed this in the revised ms.
  7. Page 13. We added a note (to the layout editor) for the presentation of the pterygial formula; however, the problematic presentation still is there. To avoid future problems, the formula was prepared as a tiff text-figure to insert in the paragraph.

  1. 13. All references were highlighted in the submitted ms. Unfortunately, [32–38] and several others lost the blue color during the uploading of the ms. This has been corrected and double-checked in all references.

Reviewer 2 Report

The manuscript is very well written and presented in a straight forward style. This is a well documented description of new important Triassic fishes that greatly expand our knowledge about Norian fish divesity, also providing new information about the paleoenvironment of the uppermost Norian of Saxony.

The descriptions of the taxa are fully informative and the pertinent features are well illustrated. The rational for the nomenclatural decision about the taxa are well founded and reasonable. The anatomical features of the described specimens are thoroughly discussed and documented. The authors present much new information and important new interpretations that will be widely cited now and in the future especially in further discussions of the history of Triassic teleosteomorphs.

Overall, I think that this is a very good manuscript and I really hope to see it published soon. 

I only have a minor question concerning the use of the term "miniaturized" when referring to taxa measuring more than 50 mm. Wietzman & Vari (1988 - Proceedings of the Biological Society of Washington, 101: 444-465) considered miniaturized fishes the diminutive species under 26 mm SL. Such a definition was also followed by Hanken & Wake (1993; Annual Review of Ecology and Systematics) in the review about miniaturization of body size. Perhaps, the use of the term "diminutive" or just "small-sized" would be more appropriate.

Author Response

We thank the reviewer for the time and effort spent reviewing the ms. and his positive words concerning the content of the ms.

Rev. 2 has one comment concerning the presentation of small body-sized fishes under the name “miniature” fishes. We agree with Rev. 2 about the strict definition of “miniature fishes” under the terms of Weitzmann and Vari (1988) that has remained unchanged with 2.6 mm standard length (SL) as a landmark for fishes sexually mature at that length or earlier and often having paedomorphic or reductive morphological traits. We are unable to know when a fossil fish has reached sexual maturity, and in addition, the SL is difficult to be precisely established in fishes possessing a heterocercal or hemiheterocercal caudal fin, so commonly the provided length of these fishes is the total length. Considering that the comment is important, we have added additional information and comments concerning fossil fishes in the section (3.1.5) of Miniaturization. We have kept the word “miniaturization” in the subheading as a way to call attention to the small body length of Triassic fishes, and the differences are discussed in this section. The wording was also slightly modified in section 4 (Conclusions: fourth paragraph, that refers to small body-sized fishes).

Reviewer 3 Report

There are a few instances where the English is a little clumsy. Pages 8 and 9  "rectangular-shaped" , perhaps better to say "rectangular in shape". page 10 "platy" isn't English at all, what do the authors mean? page 10 Triangular-shaped" maybe say "Triangular in shape"? and what is "scaly-like" on page 11? When there is a figure inserted, the text seems to lose continuity? and on page 13 the authors have said "The pterygial formula is" followed by what appears to be nonsense. Something is missing? On page 15 a whole paragraph has no italics in taxonomic names and on page 17 "only scales too fragile to be preserved and  recovered" seems to be incomplete. On page 18, "fluviatil" should be fluviatile? 

Author Response

We thank the reviewer for the time and effort spent reviewing the manuscript (ms.). Rev. 3 provided a list of comments concerning the “clumsy” use of some American English words. All of these clumsy terms have been replaced in the revised manuscript.  As it is established in Acknowledgements, the language of the ms. was checked by Dr. T.J. Meehan; a native American English-speaking colleague. He also checked the language in the revised version. We note here, as well as in comments to other reviewers and editor, that numerous changes in the layout of the ms. were automatically done by the uploading system and/or saving the submitted manuscript. These created a number of problems in the presentation of the ms., most of which were corrected one by one in this revised version, except the spaces still left between the description of some cranial bones and Figure 4 (pages 8–9), the caudal fin and Figure 5 (pages 11–12), and actinistians (Figure 8, pages 17–18) that we are unable to correct and asked the editor that this problem be corrected by their office.